# Factors Influencing Persistence of Diphtheria Immunity and Immune Response to a Booster Dose in Healthy Slovak Adults

**DOI:** 10.3390/vaccines7040139

**Published:** 2019-10-07

**Authors:** Marek Petráš, Vladimir Oleár, Milica Molitorisová, Jana Dáňová, Alexander M. Čelko, Elena Nováková, Mária Štefkovičová, Zuzana Krištúfková, Jana Malinová, Ivana Králová Lesná

**Affiliations:** 1Preventive Medicine, Third Faculty of Medicine, Charles University in Prague, 100 00 Prague, Czech Republic; jana.danova@lf3.cuni.cz (J.D.); martin.celko@lf3.cuni.cz (A.M.Č.);; 2Faculty of Healthcare, Alexander Dubček University of Trenčín, 911 50 Trenčín, Slovakia; vladimirolear@gmail.com (V.O.); tn.stefkovicova@uvzsr.sk (M.Š.); 3Faculty of Pharmacy, Comenius University in Bratislava, 832 32 Bratislava, Slovakia; milica.molitorisova@uniba.sk; 4Jessenius Faculty of Medicine in Martin, Comenius University in Bratislava, 036 01 Martin, Slovakia; elena.novakova@uniba.sk; 5Faculty of Public Health, Slovak Medical University in Bratislava, 831 01 Bratislava, Slovakia; kristufkova@gmail.com; 6Laboratory for Atherosclerosis Research, Centre for Experimental Medicine, Institute for Clinical and Experimental Medicine, 140 21 Prague, Czech Republic; ivka@ikem.cz

**Keywords:** diphtheria vaccination, seroprotection rate, smoking, statins, pre-vaccination levels, immunity persistence

## Abstract

We assessed the long-term persistence of humoral immunity against diphtheria in adults with childhood vaccination and the immunogenicity of a booster dose considering demographic, behavioural and vaccinating factors. We conducted a trial in 200 healthy Slovak adults aged 24–65 years, immunised against diphtheria in childhood and against tetanus at regular 10–15 year intervals, and receiving a dose of a tetanus-diphtheria toxoid vaccine. The response was determined by ELISA antibody concentrations of paired sera before and at 4 weeks post-vaccination. A seroprotection rate of 21% (95% confidence interval, CI 15.6–27.3%) was found in adults up to 59 years since the last vaccination with seroprotective levels of antibodies against diphtheria ≥0.1 IU/mL and a geometric mean concentration of 0.05 IU/mL. Conversely, seropositive levels ≥0.01 IU/mL were observed in 98% of adults (95% CI 95–99.5%). Booster-induced seroprotection was achieved in 78% of adults (95% CI 71.6–83.5%) clearly depending on pre-booster antibody levels correlating with age and time since the last vaccination. Moreover, only 54.2% of smokers and 53.3% of patients on statins exhibited seroprotection. Booster vaccination against diphtheria was unable to confer seroprotection in all recipients of only childhood vaccination.

## 1. Introduction

Diphtheria and tetanus vaccines are the most commonly used vaccines worldwide and generally considered successful in the immunoprophylaxis of both preventable infection diseases, also due to the relatively high vaccination rate of children and regular booster immunisation in adulthood.

While a monovalent vaccine against tetanus is usually used for booster or post-exposure vaccination after an injury, the diphtheria vaccine is available only as part of combined vaccines. The bivalent vaccine against diphtheria and tetanus indicated for both booster and primary vaccination in adults contains a reduced amount of diphtheria toxoid. Some countries require the booster dose to be repeated every 10–15 years, either with a bivalent or combined vaccine containing both toxoids. Implementation of the original program of booster immunisation against tetanus at a regular interval of 10–15 years was changed by Slovak legislation in the year of 2008 when it was expanded to include a booster dose of a bivalent vaccine against diphtheria and tetanus.

While immunisation against diphtheria has been carried out worldwide for more than 80 years, knowledge about long-term diphtheria immunity as well as results of the immune response after booster dose administration, including factors that impact it, are still not satisfactorily explored. We first focused on assessing the persistence of humoral immunity against diphtheria and, second, attempted to evaluate the immune response elicited by a bivalent vaccine containing diphtheria toxoid. For this purpose, we used data obtained from our clinical trial designed to evaluate immunogenicity of two alum-adjuvanted tetanus and diphtheria vaccines (EudraCT 2018-001604-10).

## 2. Materials and Methods

Two hundred healthy adults aged between 24 and 65 years requiring routine booster immunisation against tetanus and diphtheria were enrolled in the trial designed as a single-blind, randomised, parallel-group one like our previous trial evaluating booster-vaccination against tetanus [1]. All vaccinated subjects were selected from four Slovak outpatient clinics. Previous vaccination against tetanus carried out 10–15 years earlier was confirmed by each patient’s medical records kept by their practitioner, with confirmation of the last vaccination against diphtheria obtained in the same way. All participants had been properly and completely vaccinated with at least five doses of a combined vaccine against tetanus and diphtheria in childhood.

Subjects were randomly assigned (at a 1:1 ratio) to one of two vaccine groups stratified by sex and age to receive, from September through November 2017, either Imovax DT Adults (Sanofi Pasteur SA, Lyon, France) or Vacdite (Biodrug s.r.o., Bratislava, Slovakia). Both vaccines were administered intramuscularly into the deltoid region according to manufacturers’ recommendations.

The reference vaccine (Imovax DT Adults) was supplied in a pre-filled syringe fitted with a 16 mm needle while the test one (Vacdite) was available in an ampoule. As the needle length of the pre-filled syringe was shorter than generally recommended [2], two clinical sites were equipped with 16 mm needles while the other two with needles of the recommended minimal length (25 mm) for administration of the ampouled vaccine. Single blinding was performed by simple overlapping of subject’s eyes with a mask during the administration of vaccine in order to reduce the bias of adverse events reports dependently of the used vaccine.

The immune response was assessed by serological analysis of paired serum samples obtained from each subject before and 4 weeks after vaccination. The samples were collected, stored at below −20 °C and measured with the ELISA method using the following kits: VaccZyme Anti-Tetanus Toxoid IgG Enzyme Immunoassay with a reported measuring range of 0.01–7.0 IU/mL for tetanus antibodies and VaccZyme Anti-Diphtheria Toxoid Ig Enzyme Immunoassay with a measuring range of 0.012–3.0 IU/mL for diphtheria antibodies (Binding Site Group Ltd., Birmingham, UK). Antibody concentrations were determined in one central laboratory (Institute of Microbiology and Immunology, Martin, Slovakia) and maximum accuracy was ensured by measurement of paired samples in one run. The potential bias of serological results dependently of investigated factors was completely excluded because the laboratory staff determining antibody concentrations was blinded to subject identity.

This clinical trial was conducted in compliance with the rules of Good Clinical Practice and approved by the Ethics Committee of the Region of Trenčín and the State Institute for Drug Control in Slovakia. All subjects expressed their voluntary participation in the study by signed informed consent.

Primary seroconversion rates (SCR4) for booster response were adopted from the World Health Organization’s (WHO) requirements for post-vaccination immunogenicity comparing two vaccines [3], defined as the proportion of subjects with a four-fold increase in antibody levels before and after vaccination if attaining ≥0.4 IU/mL. In our study, seroprotection rates at levels ≥0.1 IU/mL (SPR01) and ≥1.0 IU/mL (SPR1) were considered to confer full and long-term protection against diphtheria, respectively. Immunogenicity was also assessed using geometric mean concentrations (GMCs) of diphtheria antibodies.

As antibody levels after log-transformation exhibited normal distribution, the parametric t-Student tests or one-way ANOVA test could be used for comparison of GMCs among the analysed groups. The proportions including rates were statistically evaluated with Fisher’s exact test. The sample size of 200 subjects was based on the null hypothesis investigating inferiority of the SCR4 of the test vaccine against the reference one with a delta margin of 10%.

Logistic regression was used to assess any potential association of predictors with the seroprotection rate. The demographic and behavioural characteristics were selected as possible factors influencing persistence of protective levels. Factors of immunisation were added to investigate their potential association with the booster-induced seroprotection rate. All predictors including sex and study vaccine are reported in Table 1, where they are divided into two groups according to dichotomous values or according to the median of continuous variables. The sample size was justified for logistic regression using up to 10 covariates [4]. McFadden’s R squared of this approach was >0.2, including a *p* value < 0.0001, indicating a sufficient predictive ability of this model for the selected predictors. The association was evaluated with the odds ratio mutually adjusted for all selected predictors (aOR), including a 95% confidence interval (95% CI). All tests were two-tailed, and the level of significance was set at 0.05. Statistical analyses and logistic regression were performed using Prism 8 (GraphPad Software, Inc., San Diego, CA, USA) and STATA version 15.1 software (StatCorp, Lakeway Drive, TX, USA), respectively.

## 3. Results

Before booster immunisation, there were only 42 subjects with diphtheria antibody concentrations ≥0.1 IU/mL, i.e., 21% (95% CI 15.6–27.3%) with none having antibody concentrations ≥1.0 IU/mL. The pre-booster geometric mean concentration was 0.05 IU/mL (95% CI 0.05–0.06 IU/mL) and did not vary depending on the investigated factors (Table 2).

Persistence of the SPR01 was not dependent on any of the investigated factors except concomitant medication (Table 3). Although the study was conducted in healthy adults, it was not possible to exclude subjects with concomitant diseases not constituting a contraindication to both vaccinations, such as hypertensive disease (17.5%), impaired lipoprotein metabolism (7.5%), impaired thyroid function (4.5%) and other conditions (14.5%). A higher chance of maintaining protective levels ≥0.1 IU/mL was observed in subjects without concomitant treatment, i.e., an aOR of 2.86 (95% CI 1.27–6.45). However, the concomitant treatment had no impact on the persistence of the pre-booster GMCs (Table 2).

Even if smoking was not associated with a reduction of protective levels (*p* = 0.06), they were more often observed in non-smokers (22.7%; 95% CI 16.8–29.6%) than in smokers (8.3%; 95% CI 1.0–27%). No other association between the explored factors and duration of seroprotective levels was found (Table 3).

The immune response to a single booster dose of bivalent vaccines increased both tetanus and diphtheria antibodies (*p* < 0.0001). Although the post-vaccination response to the tetanus vaccine was much stronger than that against diphtheria, both vaccines did not differ in the SCR4 achievement and induction of the GMCs of both antibodies. The SCR4 was lower for both diphtheria vaccines (39%; 95% CI 29.4–49.3% for Vacdite and 30%; 95% CI 21.2–40% for Imovax DT Adult) than that for both tetanus vaccines (75%; 95% CI 65.3–83.1% for Vacdite and 67%; 95% CI 56.9–76.1% for Imovax DT Adult). Even if most of the subjects (78%; 95% CI 71.6–83.5%) responded with protective levels of diphtheria antibodies, levels ≥1.0 IU/mL were seen in only 23.5% (95% CI 17.8–30.0%).

The immune response elicited by both study combined vaccines were independent of the vaccine since the seroconversion or seroprotection rates difference between both vaccines was always below 10% including the lower limit of a 95% confidence interval. The same results were confirmed by the non-inferior GMCs ratio of both vaccines.

The booster-induced immune response was influenced by the pre-booster levels of diphtheria antibodies, as demonstrated by an aOR of 7.5 (95% CI 2.9–19.4) for achieving SPR01 in subjects with levels >0.05 IU/mL compared to those with levels ≤0.05 IU/mL as well as a *p* value < 0.0001 for GMCs between both groups (Table 2).

While the pre-booster levels of diphtheria antibodies did not correlate with either age or time since the last vaccination, the booster-induced GMCs was higher in younger ≤43.3 years than in older subjects (*p* < 0.0001) and in those with the last vaccination ≤33.9 years earlier than later (*p* <0.0001) in conformity with the booster-induced GMCs for levels higher or lower than 0.05 IU/mL (Table 3).

The booster-elicited seroprotection rates and GMCs of diphtheria antibodies were furthermore investigated for quartile-stratified predictors, i.e., pre-booster levels and time since the last vaccination (Figure 1). At pre-booster levels ≥0.09 IU/mL, single-dose vaccination increased SPR01 to 95.5% (95% CI 77.2–99.9%) and SPR1 to 34.7% (95% CI 21.7–49.6%) achieving the highest GMCs (0.54 IU/mL; 95% CI 0.40–0.73 IU/mL). Conversely, only 56% and 8% of subjects with pre-booster levels of 0.01–0.03 IU/mL had protective levels ≥0.1 IU/mL and ≥1.0 IU/mL, respectively. Moreover, their booster-induced GMCs of diphtheria antibodies were low, i.e., 0.14 IU/mL (95% CI 0.09–0.21 IU/mL).

Subjects immunised 10 to 13 years earlier achieved seroprotective levels more often (86%; 95% CI 73.3–94.2%) than those with the last vaccination against diphtheria 45–59 years earlier (64%; 95% CI 49.2–77.1%). The GMCs of diphtheria antibodies consistent with the seroprotection rates were also higher in those vaccinated 10–13 years (0.43 IU/mL; *p* = 0.0127) as well as 13–34 years (0.49 IU/mL; *p* = 0.0055) earlier compared to those with a longer time since the last vaccination (0.19 IU/mL). However, the quartile-stratified period between the last and current immunisation against diphtheria did not show any impact on SPR1 (*p* = 0.0657).

The adjusted odds ratio of 6.5 (95% CI 2.0–21.2) revealed a difference in achieving seroprotective levels between smokers and non-smokers. While 81% of non-smokers responded to booster vaccination with seroprotective concentrations, the same response was found in only 54% of smokers (Table 3). The GMCs of diphtheria antibodies were also lower in smokers (0.16 IU/mL) than in non-smokers (0.33 IU/mL; *p* = 0.0125).

Although concomitant treatment did not reduce antibody levels under the protective threshold, it did contribute to lower their total concentrations (Table 2). A detailed analysis of the most frequently reported concomitant diseases showed that 15 patients with impaired lipoprotein metabolism (E78) had lower GMCs (0.12 IU/mL; 95% CI 0.07–0.21 IU/mL) than others (0.32 IU/mL; 95% CI 0.26–0.39 IU/mL). Moreover, only 53.3% (95% CI 26.2–78.8%) of them achieved seroprotective levels. The immune response induced by a booster dose was not influenced by any other factor explored.

Adverse events (AEs) were reported by a total of 53 subjects, i.e., 26.5% (95% CI 20.5–33.2%). Neither serious nor severe AEs were documented during the entire study. Most of the vaccination-related AEs, i.e., 79.2% (95% CI 65.9–89.2%) were mild. Their occurrence was extremely higher in subjects vaccinated with the 16-mm needle (86.4%; 95% CI 72.6–94.8%) compared to those receiving the vaccine with a longer needle (13.6%; 95% CI 5.2–27.4%). Common AEs were consistent with usual and expected reactions, such as redness, swelling, induration and itching at injection site, arm or shoulder pain, fatigue, headache, nausea and abdominal pain. The safety profile of vaccines showed good tolerability with no difference.

## 4. Discussion

The last decade has witnessed a re-emergence of vaccine-preventable diseases and, although occurrence of diphtheria is still not under control, the hazard of imported cases increases the potential threat to unvaccinated population as well as to adults immunised a long time ago. Our results could contribute to diphtheria control precautions, especially in emergency situations.

The persistence of humoral immunity within 10 to 59 years after the last dose of the vaccine against diphtheria did not exhibit any variability, independently of age. Although almost all of subjects (except four) had serum antibodies against diphtheria toxin above the putative lowest protection threshold of 0.01 IU/mL, only 21% of them retained seroprotective levels ≥0.1 IU/mL as those internationally considered to confer full protection. The present study results are in accordance with current knowledge obtained from other seroepidemiological studies [5,6,7,8]. The immune response elicited with one booster dose of both study vaccines against diphtheria and tetanus did not differ in the seroconversion rates or in the GMCs of both antibodies.

Interestingly, booster vaccination against diphtheria did not induce such a strong immune response as that against tetanus. While all subjects achieved seroprotective levels of tetanus antibodies after booster dose administration within 10–16 years, only 88.1% and 31.3% of subjects achieved seroprotective levels of diphtheria antibodies ≥0.1 IU/mL and ≥1.0 IU/mL, respectively. However, if adults younger than 30 years were regularly booster-immunised at an interval of 10–16 years, then 100% of them were seroprotected. Similar results were documented in other studies focused on booster immunisation of adults [9,10,11].

Our data support the recommendation of Slovak legislation to regularly administer, every 15 years, one booster dose of a combined vaccine against diphtheria and tetanus independently of the age of the adult. It is just the regular repetition of booster immunisation that can help to afford full protection of the population. Unfortunately, this study could not evaluate the effect of repeated booster immunisation against diphtheria in older adults because the law has been in force since the year of 2008 and only a negligible number of participants in this study was immunised in that year.

The antibody concentrations prior to booster immunisation suggested that not only antibody levels but, also, immunological memory may wane within a few decades.

Especially in cases where the pre-booster levels of antibodies dropped under 0.05 IU/mL, the chance to achieve protective levels of antibodies decreased by about 87%. This finding could be explained by the decrease in long-lived plasma B lymphocytes that helps to trigger a sufficient immune response after vaccine re-exposure [12]. Therefore, it is likely that, in this particular case, there could be a booster dose as a putative one with another dose required to achieve seroprotective levels.

Even if the key predictor of the booster-induced immune response is a persistent concentration of antibodies, it could be replaced by the time since the last vaccine administration as one more suitable for practice [13]. The present data showed that adults with the last diphtheria vaccination >34 years earlier were less responsive to booster immunisation since seroprotective levels were observed in only 69% of them. If the epidemiological situation markedly worsened, the time interval since the last vaccination could comfortably predict if only one, or two doses should be administered in an emergency. It was found that a second dose can improve an insufficient response induced by a booster dose and can contribute to increasing the levels above 0.1 IU/mL [13,14].

As the effect of cigarette smoking on vaccination is generally accepted [15], the finding of lower seroconversion rates and GMCs of diphtheria antibodies in smokers comes as no surprise. One can even speculate that smoking contributes to loss of diphtheria antibodies because a very small proportion of smokers (8.3%) had diphtheria antibodies at protective levels prior to booster immunisation.

While long-term concomitant treatment or diseases had no impact on achieving seroprotection, the post-vaccination GMCs of diphtheria antibodies were higher in healthy adults compared to patients, especially those on statins. We could not conclusively confirm in this trial if patients taking statins achieve worse post-booster immunogenicity since 15 patients represented too small a sample size and the power of test was lower than the usual 80%. This premise is, however, biologically plausible as numerous studies showed an ability of statins to modulate immuno-inflammatory processes [16]. One can assume that statins treatment can impact the vaccine-induced response in some way. Moreover, several studies have already documented that statins reduced the influenzae vaccine-induced response or protection [17,18,19]. Conversely, the fact why the same response to tetanus toxoid did not reveal any influence of this treatment can be explained by both regular immunisation and a stronger effect of tetanus toxoid than that of the diphtheria one.

Both vaccines were very well tolerated, and occurrence of AEs was not dependent on any of the investigated factors except one. It was confirmed that the vaccination-related AEs, especially local ones, were more frequent in subjects immunised with the shorter 16-mm needle than in those immunised with the longer one. Use of the 25 mm needle is probably associated with fewer moderate or mild local reactions after vaccination compared to the 16 mm needle as demonstrated in our study in accordance with other authors [20]. Vaccines supplied in pre-filled syringes reflect the current universal use for both children and adults which is why the 16 mm needle employed for childhood immunisation is generally accepted also for adults.

## 5. Conclusions

Booster immunisation against diphtheria in adults carried out with bivalent tetanus-diphtheria vaccines produced a protective immune response dependent on persistent levels of diphtheria antibodies prior to the booster dose. The persistence of antibody levels is the key predictor of seroprotection achievement correlating with time since the last vaccination.

Therefore, two-dose vaccination should be considered, in an emergency, in recipients with the last dose received >34 years earlier. Based on our data, we can further anticipate a higher increase in diphtheria antibodies more often in non-smokers and in adults without statin treatment. If these factors have any clinical impact on protection against diphtheria is yet to be established.

## Figures and Tables

**Figure 1 vaccines-07-00139-f001:**
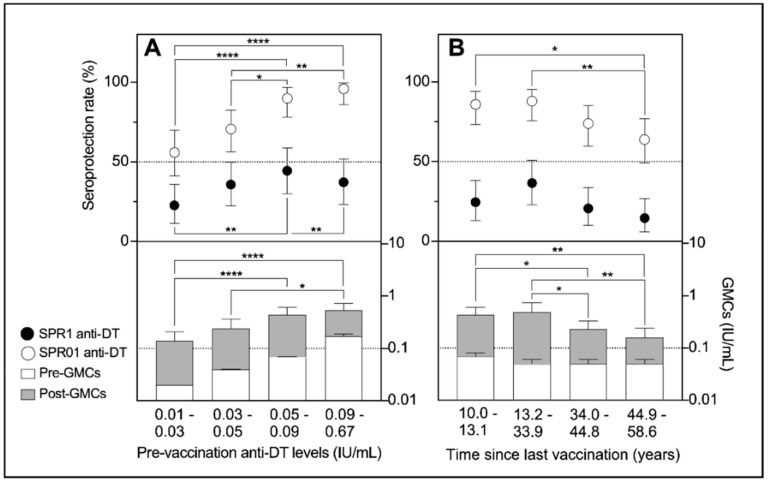
Seroprotection rates of 0.1 IU/mL and 1.0 IU/mL thresholds including the GMCs of pre- and post-booster immunisation against diphtheria in dependence on the quartile-stratified pre-booster levels (**A**) and time since the last vaccination (**B**).

**Table 1 vaccines-07-00139-t001:** Characteristics of the study population according to predictors expressed with proportions or means including the 95% confidence interval.

Predictors		N ^1^	Proportion % (95% CI)	Predictors		N	Mean (95% CI)
Smoker	Yes	24	12 (7.8–17.3)	Age (years)	≤43.3	100	32.6 (31.3–33.8)
	No	176	79.5 (73.2–84.9)		>43.3	100	53.8 (52.4–55.2)
Pre-booster concomitant medication	Yes	73	36.5 (29.8–43.6)	^2^ BMI (kg/m^2^)	≤25.2	100	22.2 (21.8–22.7)
	No	127	63.5 (56.4–70.2)		>25.2	100	29.6 (28.8–30.3)
Related adverse events	Yes	44	22 (16.5–28.4)	Time since last dose (years)	≤33.9	100	17.1 (15.5–18.7)
	No	156	78 (71.6–83.5)		>33.9	100	45.0 (43.5–46.4)
Needle length (mm)	16	150	75 (68.4–80.8)	Pre-vaccination anti-DT ^3^ GMCs (IU/mL)	≤0.05	101	0.03 (0.02–0.03)
	25	50	25 (19.2–31.6)		>0.05	99	0.11 (0.09–0.12)

^1^ N: number of subjects; ^2^ BMI: body mass index; ^3^ anti-DT anti-diphtheria toxin antibodies, ^3^ IU: international unit, ^3^ GMC: geometric mean concentration. Note: males and females as well as both vaccines at a 1:1 ratio, i.e., proportions of 50% (95% CI 42.9–57.1%).

**Table 2 vaccines-07-00139-t002:** Pre- and post-booster geometric mean concentrations (GMCs) of diphtheria antibodies, including the 95% confidence interval and *p* value between subgroups.

Predictors		GMCs (IU/mL), (95% CI)	
		Pre-Booster	Post-Booster
Sex	Male	0.06 (0.05–0.07)	0.35 (0.26–0.45)
	Female	0.05 (0.04–0.06)	0.26 (0.20–0.34)
	*p* value	NS	NS
Vaccine	Vacdite	0.06 (0.05–0.07)	0.33 (0.25–0.44)
	Imovax DT Adult	0.05 (0.04–0.05)	0.27 (0.21–0.35)
	*p* value	NS	NS
Smoker	Yes	0.04 (0.03–0.06)	0.16 (0.08–0.31)
	No	0.05 (0.05–0.06)	0.33 (0.27–0.40)
	*p* value	NS	0.0125
Needle length (mm)	16	0.05 (0.04–0.06)	0.29 (0.23–0.36)
	25	0.06 (0.05–0.07)	0.33 (0.23–0.47)
	*p* value	NS	NS
Pre-booster concomitant medication	Yes	0.06 (0.05–0.08)	0.22 (0.16–0.29)
	No	0.05 (0.04–0.05)	0.36 (0.28–0.46)
	*p* value	NS	0.0142
Related adverse events	Yes	0.04 (0.03–0.05)	0.31 (0.20–0.48)
	No	0.06 (0.05–0.06)	0.29 (0.24–0.37)
	*p* value	0.0409	NS
Age, median (years)	≤43.3	0.06 (0.05–0.07)	0.49 (0.38–0.64)
	>43.3	0.05 (0.04–0.06)	0.18 (0.14–0.23)
	*p* value	NS	<0.0001
BMI, median (kg/m^2^)	≤25.2	0.05 (0.04–0.06)	0.34 (0.26–0.45)
	>25.2	0.05 (0.04–0.06)	0.26 (0.20–0.34)
	*p* value	NS	NS
Time since last immunisation, median (years)	≤33.9	0.06 (0.05–0.07)	0.46 (0.35–0.59)
	>33.9	0.05 (0.04–0.06)	0.19 (0.15–0.25)
	*p* value	NS	<0.0001
Pre-vaccination anti-DT GMCs, median (IU/mL)	≤0.05	0.03 (0.02–0.03)	0.18 (0.14–0.24)
	>0.05	0.11 (0.09–0.12)	0.49 (0.39–0.61)
	*p* value	<0.0001	<0.0001

Note: For abbreviations, see Table 1; NS: not significant; *p* value calculated with the unpaired *t*-test.

**Table 3 vaccines-07-00139-t003:** Seroprotection rate with a threshold of 0.1 IU/mL pre- and post-booster immunisation against diphtheria, crude and mutually adjusted odds ratios (cORs, aORs) including the 95% confidence interval.

Predictors		Pre-Booster				Post-Booster			
		Rate %	cOR	aOR	P ^1^	Rate %	cOR	aOR	P ^1^
Vaccine	I ^3^	NA ^2^				75 (65–83)	1	1	NS
	V ^4^					81 (72–88)	1.4 (0.7–2.8)	0.8 (0.3–2.1)	
Sex	M ^5^	24 (16–34)	1	1	NS	63 (53–72)	1	1	NS
	F ^6^	18 (11–27)	0.7 (0.4–1.4)	0.6 (0.3–1.2)		79 (70–87)	2.2 (1.2–4.2)	0.8 (0.3–1.9)	
Smoker	Yes	8.3 (1–27)	1	1	NS	54 (33–74)	1	1	0.002
	No	23 (17–30)	3.2 (0.7–14)	4.4 (0.9–21)		81 (75–87)	3.7 (1.5–8.9)	6.5 (2.0–21)	
Pre-booster CM ^7^	Yes	17 (11–24)	1	1	0.01	71 (60–81)	1	1	NS
	No	29 (19–41)	2.0 (1.0–4.0)	2.9 (1.3–6.5)		80 (72–87)	1.7 (0.9–3.2)	1.3 (0.5–3.2)	
Age (years)	≤43.3	21 (14–30)	1	1	NS	87 (79–93)	1	1	NS
	>43.3	21 (14–30)	1.0 (0.5–2.0)	1.0 (0,4–2.9)		69 (59–78)	0.3 (0.2–0.7)	0.7 (0.2–2.7)	
BMI (kg/m^2^)	≤25.2	19 (12–28)	1	1	NS	77 (68–85)	1	1	NS
	>25.2	23 (15–33)	1.3 (0.6–2.5)	1.0 (0.5–2.1)		79 (70–87)	1.1 (0.6–2.2)	0.8 (0.3–1.9)	
Time since last dose (years)	≤33.9	23 (15–33)	1	1	NS	87 (79–93)	1	1	NS
	>33.9	19 (12–28)	0.8 (0.4–1.6)	0.5 (0.2–1.4)		69 (59–78)	0.3 (0.2–0.7)	0.3 (0.1–1.0)	
Related adverse events	Yes	NA				77 (62–89)	1	1	NS
	No					78 (71–84)	1.1 (0.5–2.4)	0.7 (0.3–2.1)	
Pre ^8^ anti-DT GMCs (IU/mL)	≤0.05	NA				63 (53–73)	1	1	<0.001
	>0.05					93 (86–97)	7.6 (3.2–18)	7.5 (2.9–19)	
Needle length (mm)	16	NA				84 (71–93)	1	1	NS
	25					76 (68–83)	0.6 (0.3–1.4)	1.4 (0.4–4.6)	

^1^*p* value; ^2^ NA: not applicable; ^3^ I: Imovax DT Adult; ^4^ V: Vacdite; ^5^ M: Male; ^6^ F: Female; ^7^ CM: concomitant medication; ^8^ Pre-vaccination; 9 NS: not significant. Note: For abbreviations, see Table 1.

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
