# Peer review of "Factors Influencing Persistence of Diphtheria Immunity and Immune Response to a Booster Dose in Healthy Slovak Adults"

_vaccines, 2019, doi:10.3390/vaccines7040139_

Round 1

Reviewer 1 Report

The manuscript "Factors influencing persistence of diphtheriae 3 immunity and immune response to a booster dose in healthy Slovak" adults by Marek Petráš et al. faces the relevant topic of the persistence of post-immunization diphtheria antibodies, by analyzing  one cohort of adults randomized to receive a booster from  either one of the two tetanus/diphtheria bivalent vaccines chosen. In particular, the current manuscript is addressed to investigate the diphtheria response, having the response to tetanus already been described in another distinct paper in 2018.

The final message of the study is the recommendation to administer at least 2 boosters in people immunized ≥34 years, allowing to by-pass the need of a previous analysis of antibody levels. However, despite the interest of the topic, some aspects need to be addressed to improve the paper and make it publishable.

Authors use indifferently the words “diphtheria” and “diphtheriae” throughout the manuscript, but they should use the word “diphtheria” to indicate the disease and limit the use of the “diphtheriae” to the indication of the causative agents of diphtheria, the Clostridium diphtheriae.

Line 42:  “.. reduced amount of diphtheriae and/or tetanus toxoids” appears misunderstanding. In fact, the adult formulation of Td always uses an amount of tetanus toxoid higher than the amount of diphtheria toxoid (which is approximately one tenth compared to tetanus), considering its high reactogenicity in adults. (from WHO: Diphtheria and Tetanus toxoid combination: DT vaccine used for primary immunisation and boosting in children contains 6.7-25Lf of diphtheria toxoid and 5 – 7.5 Lf of tetanus toxoid per dose. An adult combination, Td, is used for boosting and primary immunisation in adolescents and adults and contains a lower dose of diphtheria (less than 2 Lf/dose) but a similar dose of tetanus toxoid.)

Table 1: In the text is reported that 200 subjects were studied, but the sum of smokers and non-smokers is lower than 200 (183).

Table 3: It is very difficult to follow. A grid may probable help the reader.

In tables and Figures, I suggest using: “anti-Dt” to abbreviate “anti-diphtheria toxin antibody” since DT is widely used to abbreviate Diphtheria-Tetanus.

Line 120: I do not understand what authors want to underline with the sentence: “The geometric mean concentrations of 0.05 IU/mL (95% CI 0.05–0.06 IU/mL) were low”.

Lines 133-135: I do not understand the message of the sentence. Do they mean that the two parameters (the time since the last immunisation against diphtheria (range, 10–59 years) as well as concomitant treatment) seem to have a significant impact on the post-immunization increase (and not persistence) of GMCs ?. It could be useful to report in one table the range of values and not only GMCs with CI, as well as an additional table with the results of logistic regression, in order to verify the significance of post-immunization levels for these two parameters in a multivariate analysis.

Several vaccines use CRM197 as a carrier protein for polysaccharides. Antibodies elicited against this carrier protein are indistinguishable from those elicited by anti-diphtheria vaccines by current ELISA methods. Results obtained evaluating by ELISA the response to booster doses of DT vaccines in dependence of the temporal interval from the primary immunization with DT vaccines may be influenced by previous vaccinations with conjugated polysaccharides. Did authors investigate for previous conjugate polysaccharide vaccinations, i.e. anti-pneumococcus (7-13 valent) or anti-meningococcal (1-4 valent) vaccines?

In the Results section, lines 160-163, it is reported that the post-immunization antibody levels are influenced by the pre-immunization levels and in Table 2 and Fig. 1, A, it is possible to observe that this association is direct, with lower pre-immunization antibody levels associated to a lower response and vice versa; this is not in line with previous papers reporting an indirect correlation observed in different vaccination models and the different results should be discussed (Adv Immunol, 8 (1968), pp. 81-127, Vaccine, 32 (2014), pp. 4220-4227 Annu Rev Immunol, 18 (2000), pp. 709-737;  Vaccine 36 (2018), pp 6718-6725).

In the discussion, authors claim that “a two-doses vaccination should be considered, in an emergency, in recipients with the last dose received >34 years earlier”. Since single anti-diphteria toxin vaccines are not commercially available, the proposed two doses vaccination (at how many days of interval?) will include a two doses additional anti-tetanus vaccination. Do authors think that it will be safe? What they consider an emergency? Please specify.

Author Response

Replies to comments and suggestions of Reviewer No 1

The manuscript "Factors influencing persistence of diphtheriae 3 immunity and immune response to a booster dose in healthy Slovak" adults by Marek Petráš et al. faces the relevant topic of the persistence of post-immunization diphtheria antibodies, by analyzing  one cohort of adults randomized to receive a booster from  either one of the two tetanus/diphtheria bivalent vaccines chosen. In particular, the current manuscript is addressed to investigate the diphtheria response, having the response to tetanus already been described in another distinct paper in 2018.

Reply:

The current manuscript is focused on the immune response to diphtheriae toxoid as a component of two commercial bivalent vaccines against diphtheria and tetanus that were used in this reported trial. Another paper, published in 2018, evaluated data from other trial that was conducted by administration of monovalent vaccines against tetanus in the Czech population.

Design of both studies was similar, therefore we reported: “ … the trial designed as a single-blind, randomised, parallel-group one like our previous trial evaluating booster-vaccination against tetanus [1].

The final message of the study is the recommendation to administer at least 2 boosters in people immunized ≥34 years, allowing to by-pass the need of a previous analysis of antibody levels. However, despite the interest of the topic, some aspects need to be addressed to improve the paper and make it publishable.

Reply:

The goal of our trial was not to assess any potential improvement of the immune response to diphtheriae toxoid depending on the second dose. Therefore, we are unable to make any recommendations and only suggested that two-dose vaccination should be considered in those vaccinated more than 34 years earlier.

Authors use indifferently the words “diphtheria” and “diphtheriae” throughout the manuscript, but they should use the word “diphtheria” to indicate the disease and limit the use of the “diphtheriae” to the indication of the causative agents of diphtheria, the Clostridium diphtheriae.

Reply:

We agree and the word “diphtheriae” has been replaced by “diphtheria” throughout the manuscript.

Line 42:  “.. reduced amount of diphtheriae and/or tetanus toxoids” appears misunderstanding. In fact, the adult formulation of Td always uses an amount of tetanus toxoid higher than the amount of diphtheria toxoid (which is approximately one tenth compared to tetanus), considering its high reactogenicity in adults. (from WHO: Diphtheria and Tetanus toxoid combination: DT vaccine used for primary immunisation and boosting in children contains 6.7-25Lf of diphtheria toxoid and 5 – 7.5 Lf of tetanus toxoid per dose. An adult combination, Td, is used for boosting and primary immunisation in adolescents and adults and contains a lower dose of diphtheria (less than 2 Lf/dose) but a similar dose of tetanus toxoid.)

Reply:

We agree and the sentence has been re-worded accordingly to read: “The bivalent vaccine against diphtheria and tetanus indicated for both booster and primary vaccination in adults contains a reduced amount of diphtheria toxoid.”

Table 1: In the text is reported that 200 subjects were studied, but the sum of smokers and non-smokers is lower than 200 (183).

Reply:

We agree and the wrong number has been changed to 176.

Table 3: It is very difficult to follow. A grid may probable help the reader.

Reply:

We agree and have separated the rows.

In tables and Figures, I suggest using: “anti-Dt” to abbreviate “anti-diphtheria toxin antibody” since DT is widely used to abbreviate Diphtheria-Tetanus.

Reply:

It is true that DT is usually used as an abbreviation for a vaccine against diphtheria and tetanus. Conversely, anti-DT is often used as an abbreviation for antibodies against diphtheria toxin. The "DT" abbreviation was also used to refer to asdiphtheria toxin in the article that you referred to below (Vaccine 36 (2018), pp 6718-6725). Therefore, we think that "anti-DT" would be more appropriate for the reader than “anti-Dt”.

Line 120: I do not understand what authors want to underline with the sentence: “The geometric mean concentrations of 0.05 IU/mL (95% CI 0.05–0.06 IU/mL) were low”.

Reply:

We agree and the wording has been changed as follows: “The pre-booster geometric mean concentration was 0.05 IU/mL (95% CI 0.05–0.06 IU/mL) and did not vary depending on the investigated factors (Table 2).”

Lines 133-135: I do not understand the message of the sentence. Do they mean that the two parameters (the time since the last immunisation against diphtheria (range, 10–59 years) as well as concomitant treatment) seem to have a significant impact on the post-immunization increase (and not persistence) of GMCs ?. It could be useful to report in one table the range of values and not only GMCs with CI, as well as an additional table with the results of logistic regression, in order to verify the significance of post-immunization levels for these two parameters in a multivariate analysis.

Reply:

We agree and the paragraph has been re-worded as follows:

“Persistence of the SPR01 was not dependent on any of the investigated factors except concomitant medication (Table 3). Although the study was conducted in healthy adults, it was not possible to exclude subjects with concomitant diseases not constituting a contraindication to both vaccinations such as hypertensive disease (17.5%), impaired lipoprotein metabolism (7.5%), impaired thyroid function (4.5%) and other conditions (14.5%). A higher chance of maintaining protective levels ≥0.1 IU/mL was observed in subjects without concomitant treatment, i.e., an aOR of 2.86 (95% CI 1.27–6.45). However, concomitant treatment had no impact on the persistence of the pre-booster GMCs (Table 2).”

Note: We decided to apply multivariant linear regression of post-GMCs as a dependent variable on the investigated predictors. The results of linear and logistic regression of both dependent variables, i.e., the post-GMCs and seroprotection rate, respectively, were similar. Therefore, we consider results from logistic regression to be sufficient for the reader.

Several vaccines use CRM197 as a carrier protein for polysaccharides. Antibodies elicited against this carrier protein are indistinguishable from those elicited by anti-diphtheria vaccines by current ELISA methods. Results obtained evaluating by ELISA the response to booster doses of DT vaccines in dependence of the temporal interval from the primary immunization with DT vaccines may be influenced by previous vaccinations with conjugated polysaccharides. Did authors investigate for previous conjugate polysaccharide vaccinations, i.e. anti-pneumococcus (7-13 valent) or anti-meningococcal (1-4 valent) vaccines?

Reply:

In Slovakia, there are currently five conjugated vaccines available for adults: Prevenar 13 (since 2011), Menveo (since 2010), Nimenrix (since 2012), Neisvac (since 2009) and Menjugate (since 2003). The carrier proteins used in these vaccines are tetanus toxoid (Nimenrix, Neisvac) or the mutant diphtheria protein CRM197 (Prevenar 13, Menveo and Menjugate). Although none of these vaccines is indicated for active immunisation against tetanus or diphtheria, all of them contribute to humoral immunity specific to tetanus or diphtheria. However, the contribution rate of carrier proteins has been markedly lower than that of target vaccines (Ladhani SN, Andrews NJ, Waight P, Hallis B, Matheson M, England A, Findlow H, Bai X, Borrow R, Burbidge P, Pearce E, Goldblatt D, Miller E. Interchangeability of meningococcal group C conjugate vaccines with different carrier proteins in the United Kingdom infant immunisation schedule. Vaccine. 2015 Jan 29;33(5):648-55.").

Despite this, no subject enrolled in our trial has medical records of these vaccines within the last 10 years. Moreover, the serological results were in line with this documentation.

Given the above, we did not consider this issue of any relevance in our manuscript.

In the Results section, lines 160-163, it is reported that the post-immunization antibody levels are influenced by the pre-immunization levels and in Table 2 and Fig. 1, A, it is possible to observe that this association is direct, with lower pre-immunization antibody levels associated to a lower response and vice versa; this is not in line with previous papers reporting an indirect correlation observed in different vaccination models and the different results should be discussed (Adv Immunol, 8 (1968), pp. 81-127, Vaccine, 32 (2014), pp. 4220-4227 Annu Rev Immunol, 18 (2000), pp. 709-737;  Vaccine 36 (2018), pp 6718-6725).

Reply:

Ferlito C, Biselli R, Mariotti S, von Hunolstein C, et al. Tetanus-diphtheria vaccination in adults: the long-term persistence of antibodies is not dependent on polyclonal B-cell activation and the defective response to diphtheria toxoid re-vaccination is associated to HLADRB1∗ Vaccine. 2018 Oct 29;36(45):6718-6725.

In this study, the mean number of years since the last Td vaccination was 10 years. The calculated mean duration of protection (≥0.1 IU/mL) was about 20 years for diphtheria. The conclusion is line with our finding. The relationship between the pre- and post-vaccination levels was not evaluated.

Heyman B. Regulation of antibody responses via antibodies, complement, and Fc receptors. Annu Rev Immunol. 2000;18:709-37.

This study was conducted in receptor-deficient mice and the authors did not assess Td vaccination, hence, we did not consider their results and conclusions appropriate for comparison with our findings.

Idoko OT, Okolo SN, Plikaytis B, Akinsola A, et al. The impact of pre-existing antibody on subsequent immune responses to meningococcal A-containing vaccines. Vaccine. 2014 Jul 16;32(33):4220-7.

The study enrolled subjects younger than 29 years (children, adolescents and young adults). The authors investigated meningococcal vaccination and assessed the immunisation rate with vaccination response rate (VRR), i.e., the proportion of subjects achieving a 4-fold increase in post-vaccination levels compared to the pre-vaccination ones. This parameter exhibits different characteristics in comparison to the seroprotection rate with a cut-off value. We also investigated the VRR but decided not to include it in our manuscript because its clinical interpretation is not obviously clear.

Uhr JW, Möller G. Regulatory effect of antibody on the immune response. Adv Immunol. 1968;8:81-127.

Unfortunately, this article was not available to us. The publisher´s summary suggested that the article generally focused on the mechanism of the immune response, therefore, we are not sure that it would be useful to discuss it in our manuscript.

In the discussion, authors claim that “a two-doses vaccination should be considered, in an emergency, in recipients with the last dose received >34 years earlier”. Since single anti-diphteria toxin vaccines are not commercially available, the proposed two doses vaccination (at how many days of interval?) will include a two doses additional anti-tetanus vaccination. Do authors think that it will be safe? What they consider an emergency? Please specify.

Reply:

The goal of our trial was not to assess any potential improvement of the immune response to diphtheria toxoid depending on the second dose. We refer to the article whose authors investigated the immune response not only of one booster dose but also two booster doses more than 8 weeks apart (Hasselhorn, H.M.; Nübling, M.; Tiller, F.W.; Hofmann, F. Diphtheria booster immunization for adults. Dtsch Med Wochenschr. 1997, 122, 281-286.). The safety of both booster vaccinations was similar.

An emergency could be broadly defined as that currently encountered with measles. The vaccination rate slowly declines and the risk of recurrent vaccine-preventable infections is very high.

Reviewer 2 Report

The last decade has witnessed a re-emergence of vaccine-preventable diseases such as diphtheria. The present manuscript deals with the hazard of increases threat to unvaccinated population as well as to adults immunised a long time ago.

In this review, the authors discuss factors that could contribute to diphtheria control precautions, especially in emergency situations. They indicated that the persistence of humoral immunity within 10 to 59 years after the last dose of the vaccine against diphtheria, did not exhibit any variability, independently of age. All subjects (except four) had serum antibodies against diphtheria toxin above the putative lowest protection threshold of 0.01 IU/mL and only 21% of them retained seroprotective levels ≥0.1 IU/mL.  

They also showed that their results concurred with current knowledge obtained from other seroepidemiological studies. This means that the immune response elicited with one booster dose of vaccines against diphtheria and tetanus, did not differ in the seroconversion rates or in the GMCs of both antibodies.

Also, interestingly, the authors found that booster vaccination against diphtheria did not induce a strong immune response, as that against tetanus. While all subjects achieved seroprotective levels of tetanus antibodies after booster dose administration within 10–16 years, only 88.1% and 31.3% of subjects achieved seroprotective levels of diphtheria antibodies ≥0.1 IU/mL and ≥1.0 IU/mL, respectively.

However, the authors found that if adults younger than 30 years were regularly booster-immunised at an interval of 10–16 years, then 100% of them were sero-protected. In addition, the main conclusion of the authors was that the antibody concentrations prior to booster immunisation suggested that not only antibody levels but, also, immunological memory may wane within a few decades.

The present data showed that adults with the last diphtheria vaccination >34 years earlier were less responsive to booster immunisation since seroprotective levels were observed in only 69% of them.

If the epidemiological situation markedly worsened, the time interval since the last vaccination could comfortably predict if only one, or two doses should be administered in an emergency. It was found that a second dose can improve an insufficient response induced by a booster dose and can contribute to increasing the levels above 0.1 IU/mL.

To sum it up, the authors conclude that booster immunisation against diphtheria in adults carried out with bivalent tetanus-diphtheria vaccines can produce a protective immune response dependent on persistent levels of the diphtheria antibodies correlating with time since the last vaccination. Therefore, two-dose vaccination should be considered, in an emergency, in recipients with the last dose received >34 years earlier.

This review is well written and the immunology and vaccine used against diphtheria are well highlighted and explained. Just a minor comments, please check the English spelling carefully.

Author Response

Reply:

Thank you for your most valuable comments; we have checked the English spelling.

Reviewer 3 Report

I read with interest this manuscript. The authors reported the results of a clinical trial related to the booster doses of two bivalent Diptheria and Tetanus vaccines. The paper is interesting but there are some points which need to be clarified by the authors.

The authors reported that this study was a single-blind clinical trial. This point should be expanded in the methods section- and more details should be included. Further, in the discussion section, the authors could discuss the potential impact of the single blindness on the internal validity of the study (bias). What is the response rate? The discussion is too long and lacks a final and inclusive conclusion. 

Author Response

I read with interest this manuscript. The authors reported the results of a clinical trial related to the booster doses of two bivalent Diptheria and Tetanus vaccines. The paper is interesting but there are some points which need to be clarified by the authors.

The authors reported that this study was a single-blind clinical trial. This point should be expanded in the methods section- and more details should be included.

Reply:

We agree; the text has been expended as follows:

“Single blinding was performed by simple overlapping of subject’s eyes with a mask during the administration of vaccine in order to reduce the bias of adverse events reports dependently of the used vaccine.”

“The potential bias of serological results dependently of investigated factors was completely excluded because the laboratory staff determining antibody concentrations was blinded to subject identity.”

Further, in the discussion section, the authors could discuss the potential impact of the single blindness on the internal validity of the study (bias).

Reply:

We agree. The issue is now discussed in more detail in the “Material and Methods” section. See the reply above.

What is the response rate?

Reply:

The investigated response rate in our study was seroprotection rate at levels ≥0.1 U/mL (considered to confer full protection).

The discussion is too long and lacks a final and inclusive conclusion. 

Reply:

We agree and a “Conclusion” section has been added, see below.

“Booster immunisation against diphtheria in adults carried out with bivalent tetanus-diphtheria vaccines produced a protective immune response dependent on persistent levels of diphtheria antibodies prior to the booster dose. The persistence of antibody levels is the key predictor of seroprotection achievement correlating with time since the last vaccination.

Therefore, two-dose vaccination should be considered, in an emergency, in recipients with the last dose received >34 years earlier. Based on our data, we can further anticipate a higher increase in diphtheria antibodies more often in non-smokers and in adults without statin treatment. If these factors have any clinical impact on protection against diphtheria is yet to be established.”

Round 2

Reviewer 1 Report

Authors answered to the criticisms.

Reviewer 3 Report

All my concerns have been addressed by the authors.